# Association between Musculoskeletal Injuries and Depressive Symptoms among Athletes: A Systematic Review

**DOI:** 10.3390/ijerph20126130

**Published:** 2023-06-15

**Authors:** Priscila Marconcin, Ana Lúcia Silva, Fábio Flôres, Alexandre Nunes, Joana Filipa Lourenço, Miguel Peralta, Beatriz Minghelli

**Affiliations:** 1KinesioLab, Research Unit in Human Movement, Piaget Institute, Av. João Paulo II, lote 544, 2º andar, 1950-157 Lisboa, Portugal; 2Centro Interdisciplinar de Performance Humana (CIPER), Faculdade de Motricidade Humana, Universidade de Lisboa, Cruz-Quebrada-Dafundo, 1499-002 Lisboa, Portugal; 3Research Center in Sports Performance, Recreation, Innovation and Technology (SPRINT), 4960-320 Melgaço, Portugal; 4Instituto de Saúde Ambiental (ISAMB), Faculdade de Medicina, Universidade de Lisboa, 1495-751 Lisboa, Portugal

**Keywords:** mental health, damages, competition

## Abstract

Musculoskeletal (MSK) injuries have a significant physical and psychological influence on an athlete’s life. A systematic review of prospective cohort, cross-sectional, and case-control studies was undertaken in this study to analyze the association between MSK injuries with depressive symptoms in athletes. We searched on PubMed, Web of Science, and Scopus, with data inception to 15 February 2023. The methodological quality was assessed using the Newcastle-Ottawa Scale (NOS). Of the 3677 potential studies only nine were included. These studies showed a bidirectional association between MSK injuries and depressive symptoms. Athletes with MSK injuries had higher levels of depressive symptoms, which raises the likelihood of experiencing depression in the future. Women athletes had higher levels of depressive symptoms compared with men. The presence of depressive symptoms is a significant predictor of disability in athletes. Our findings suggest that coaches should be more aware of depressive symptoms, in order to prevent MSK injuries, and to monitor athletes following an MSK injury.

## 1. Introduction

Despite the fact that the literature has extensively studied the health and psychosocial benefits of sports involvement, the research on sport-related injuries and their association with depressive symptoms has only recently attracted growing interest and attention [1]. A sports-related injury can be defined as any physical damage sustained by an athlete as a result of participating in sports. These injuries can occur during competitions or training sessions, regardless of medical attention or time-offs sport-related [2]. Musculoskeletal (MSK) injuries are the most common type of sport-related injuries and often predispose athletes to further injuries over time [1]. MSK injuries can involve tissue-level or structural damage to the bone, tendon, muscle, ligaments, or a combination of these structures [3]. Hence, the types and locations of those injuries vary depending on the sport. According to the National, Collegiate Athletic Association estimates that over a million injuries have occurred among college student-athletes in the last years [4]. Ankle sprain, for example, had an incidence of 1.00 per 10,000 among men’s football players, wrestlers, and ice hockey athletes [5]. Another notable example is the lateral ligament complex of the ankle, which has impacted nearly five basketball players per 10,000 [6]. Furthermore, the literature also points out that a squad of 25 football players can expect about 15 muscle injuries per season [7].

Furthermore, older players are more likely to sustain injuries than younger players, and a previous injury is a risk factor for knee, ankle, hamstring, and groin sprains [8]. Some Scandinavian studies also found that sports injuries account for approximately 19% of all acute injuries, with knee and ankle sprains being the most common [9,10].

Depression is a common mental disorder that affects 5% of the adult population and is a global cause of disability [11]. The most common depressive symptoms are the loss of energy and increased fatigue, the difficulty thinking, concentrating, or making decisions [12]. The loss of interest or pleasure in physical activities is also associated with depressive symptoms [12]. The prevalence of depression in athletes is considerable, with approximately 34% of elite-level athletes reporting depressive symptoms [13]. According to Wolanin et al. [14], 23.7% of college athletes showed clinically relevant depressive symptoms, and 6.3% had moderate-to-severe levels of depressive symptoms. The authors also found that female college athletes had a 1.8 times higher risk of having depressive symptoms compared to their male peers [14]. Furthermore, depression represents a harmful effect on sporting performance [15].

The high level of depressive symptoms among athletes can be explained by different reasons. The stress and the pressure to high-performance, the high amount of traveling during the season, and the social standing demands, are associated with players’ injuries [16]. Despite the abovementioned studies, the association between depression and MSK injuries requires further investigation to determine whether depressive symptoms are, in fact, risk factors for MSK injuries or on the other hand, if the occurrence of MSK injuries that predisposes athletes to depressive symptoms. Furthermore, more research is needed into sex-related and their association with MSK injuries, and how different sports predispose to MSK injuries and depressive symptoms.

To the best of our knowledge, numerous papers have compared MSK injuries and depressive symptoms, but no other study has attempted to extensively analyze the relationship between MSK injuries and depressive symptoms. Therefore, this systematic review aims to summarize existing studies describing the association between MSK injuries and depressive symptoms in athletes of different sports, in order to address the research question: “How are depressive symptoms related to musculoskeletal injuries in athletes?”

## 2. Materials and Methods

This systematic review protocol was registered at PROSPERO (CRD42023399334) and was conducted according to the Preferred Reporting Items for Systematic Reviews and Meta-Analyses (PRISMA) 2020 guidelines [17].

### 2.1. Literature Search Strategy

After the definition of search terms, extracted from descriptors of Medical Subject Heading (MESH), we defined the following terms: (athlete* OR athletics* OR sport*) AND (depress* OR “mental health” OR mood OR “psychological health” OR “mental function*” OR worries OR worry OR “dysthymia” OR “mental ill-health” OR affective) AND (injury* OR sprain OR strain OR tendon* OR fasciitis OR bursitis). The same terms were used for all databases.

To select manuscripts, two researchers (PM and AN) screened the articles based on titles and abstracts, followed by full-text reads. A third researcher (FF) was requested to mediate any disagreements. A thorough search was conducted on articles published until 15 February 2023, and no starting date has been set. The databases used for the search were PubMed, Web of Science, and Scopus. The search was limited to peer-reviewed studies published in English and assessing humans.

### 2.2. Study Selection

The inclusion criteria were samples composed of athletes with MSK injury or athletes who had suffered an MSK injury; reported depressive symptoms using a validated instrument or with a clinical diagnosis of depression; analyzed the association of depressive symptoms and MSK injury; a cohort, case-control or a cross-sectional study design. Exclusion criteria included articles written in languages other than English, literature reviews, non-peer-reviewed studies, or opinion articles.

### 2.3. Data Extraction

Data regarding the country where the study was conducted, design, participants’ characteristics, sport modality and level of participation, depression assessment, MSK injury type, and location and results were extracted by one author (PM). Another author (FF) performed the verification of all extracted data to ensure accuracy.

### 2.4. Methodological Quality Assessment

Two researchers (AN and ALS) independently assessed the methodological quality of the listed studies. For cohort and case-control studies, the Newcastle-Ottawa Scale (NOS) was employed [18], while a modified version of the NOS was utilized for cross-sectional investigations [19]. A final score was obtained for each study within three broad perspectives: selection of the study groups; comparability among different groups; and the ascertainment of either the exposure or outcome of interest. The results were compared and any inconsistencies were discussed until a final score was reached. A third researcher (PM) was consulted in case of no agreement between the first two researchers. Quality scores lower than 4 were rated as low-quality studies, whereas those equal to or higher than 5 were rated as high-quality studies [20].

The strength of agreement between reviewers was determined through Cohen’s kappa [21]. Interpretation of k values was established using the scale proposed by Landis and Koch [22]: 0 = poor agreement, 0.01–0.20 = slight agreement, 0.21–0.40 = fair agreement, 0.41–0.60 = moderate agreement, 0.61–0.80 = substantial agreement, and 0.81–1 = almost perfect or perfect agreement.

## 3. Results

### 3.1. Study Selection

Our search identified 3677 potential studies. We removed 1706 duplicates and 194 systematic reviews, resulting in 1777 studies analyzed by title and abstract. After excluding studies that did not meet the inclusion criteria, 77 studies were assessed for eligibility. Out of these, 41 did not analyze athletes, and 11 did not assess depressive symptoms using a validated instrument or clinical diagnosis. Furthermore, in 10 studies the injuries were not MSK, and in five studies the association between MSK injury and depressive symptoms was not explored. Thus, nine studies were included in the final analysis [23,24,25,26,27,28,29,30,31]. The PRISMA flowchart that summarized the study’s identification process is shown in Figure 1.

### 3.2. Study Characteristics

The study characteristics included in this systematic review are presented in Table 1. Two included studies were from the 1990s [25,28], one from the 2000s [23], and the other six were published from 2014 until 2020 [24,26,27,29,30,31]. Regarding location, one study was from Canada [25], five from the USA [23,26,28,29,30], and the other three from Europe [24,26,31]. Most studies were prospective cohort studies (n = 5) [23,27,28,30], three studies were cross-sectional [24,25,26] and one was a case-control study [31].

### 3.3. Participants’ Characteristics

The total sample size was 2578 athletes (men: n = 2026; women: n = 552), with ages ranging from 18 to 27 years old. Only one study looked at older athletes (case group had a mean age of 41 and the control group had a mean age of 38) [27]. Two studies omitted to provide age information [28,30]. The study with the largest sample had 958 athletes [27] followed by 540 athletes [31]. Two studies had 330 [30], and 276 athletes [28], whereas the remaining studies had less than 200 participants [23,24,25,27,29].

### 3.4. Sport Modality and Level

The majority of athletes participated in team sports, primarily American football, followed by basketball, football, volleyball, baseball, hockey, rugby, softball, and field hockey, with a minority participating in individual sports such as gymnastics, track and field, wrestling, athletics, running, fitness, and paddle. In terms of sport experience, only two studies distinguished player experience [23,31]. In the studies from Appaneal et al. [23] 71% were starters athletes and 85% were collegiate athletes. And in the Gouttebarge et al. study [31], 53.5% of football players belonged to the highest level (top league). No information regarding the participants´ levels of sports performance was mentioned in the remaining studies.

### 3.5. Depressive Symptoms Assessment

Different instruments were used to assess depressive symptoms, with the Center for Epidemiologic Studies Depression Scale (CES-D) being the most commonly used (used in four studies) [23,26,29,30]. The Profile of Mood States (POMS) was used in three studies [23,25,28]. One study used the Patient-Health-Questionnaire-2 (PHQ-2) [24], another study used the Beck Depression Inventory-II (BDI-II) [27], and the other study used the 12-item General Health Questionnaire (GHQ-12) [31].

### 3.6. Musculoskeletal Injuries

The majority of studies lacked clarity in describing the type of injury sustained by the athlete in terms of MSK injuries. Nonetheless, all studies agreed on the definition of injury as “Physical trauma that resulted in restricted (no) participation for a minimum of one week”. Three studies did not define the type of MSK injury [25,28,31]; one only looked at myofascial pain [27], while the other only looked at fractures [29], and another study investigated back pain [24] One study highlighted the most common region of injury (knee) and the most common type (sprain) [30]. Sprain injury was the most prevalent type of MSK injury in one study [26], whereas another study displayed both the location and type of the athletes’ injuries, and as the following: joints (ligament and dislocations), muscle, bone (fractures), neck (cervical spine), and low back (disc) [23]. One study looks only for fractures [29].

### 3.7. Main Results

Two studies concluded that MSK injury increases depression symptoms [23,28]. In contrast, one study [29] concluded that injured athletes showed fewer depressive symptoms compared to non-injured athletes. Two studies analyzed if depressive symptoms increased the risk of MSK injury, and concluded that depressive symptoms are risk factors for injury [25,30]. It is important to highlight that, in one study, the risk of injury in men athletes only increases when depression is combined with anxiety [26]. Furthermore, one study pointed out the prevalence of 21.7% of depression among injured athletes [26] and one study found that women had more depressive symptoms than males, although men had more serious injuries [32]. Three cross-sectional studies analyzed different perspectives concerning depression and injuries [24,25,31]. One study found that the risk for depression was a significant predictor of disability among athletes with back pain [24], while another found that depression is related to injury frequency [25], and Gouttebarge et al. [31] reported that athletes with persistent severe MSK injuries during their sports career were up to four times more likely to report symptoms of common mental disorders than those who had not suffered from severe MSK injuries. The author’s findings were supported by a large sample in a cross-sectional study.

In the case-control study, athletes with MSK injury had greater depression symptoms compared with athletes without MSK injury [27], corroborating the evidence gathered in the cohort and cross-sectional studies.

### 3.8. Studies Quality Assessment

The methodological quality based on NOS is summarized in Table 2, Table 3 and Table 4. The NOS ranged from 0 to 7 points, with a median of 4 points. Of the five cohort studies, three were considered to be high-quality [24,27,31] and two were considered low-quality investigations [28,29]. The case-control study was considered high-quality [27] and the three cross-sectional studies were considered to be low-quality studies [24,25,31]. Cohen’s kappa for the strength of agreement between reviewers concerning study quality was k = 0.62 showing substantial agreement.

## 4. Discussion

Sports participation carries a significant risk of injury for both elite and recreational athletes, as athletes must possess a high level of physical (and mental) abilities to achieve peak performance [32]. Therefore, this systematic review explores the association between MSK injuries and depressive symptoms among athletes. We analyzed nine studies that meet the inclusion criteria and found a bidirectional association between the investigated variables. It seems that an MSK injury increases the risk of depressive symptoms, agreeing with other findings [32,33,34].

Although postinjury depression scores seem to gradually decrease toward baseline scores, one study [28] found that depressive symptoms peaked one week after the injury and then declined after one-month postinjury. These results are supported by McDonald and Hardy [34] who discovered increased levels of depressive symptoms one week after the injury and a decrease over time. Surprisingly, Tomlinson et al. [29] found that injured athletes exhibited fewer depressive symptoms than non-injured athletes, however, the reduced sample and its methodological low quality might negatively bias these results. Despite these controversial findings, it is important to notice the mechanisms behind MSK injuries and the associated depressive symptoms. One explanation could be the relation between the injury recovery timeline (which prevents athlete participation in training/competitions) and the feeling of identity loss or anger caused by an injury [35], which leads to increased depressive symptoms [30,35].

The study from Belz et al. [24] explored the association between MSK injury and pain. The investigation findings showed back pain is associated with stress and depression in competitive athletes, but no significant differences were found regarding sex. It is important to highlight that MSK injury may lead athletes to increase pain status, and pain is an important predictor of depressive symptoms [36], however, depression itself is a risk factor for different types of MSK pain [37,38]. Also, three studies [25,26,30] found that depressive symptoms increased the risk of MSK injury.

Overall, experiencing depressive symptoms could lead to physiological and attention-related changes, such as heightened muscle tension and distractibility, which may increase the risk of an MSK injury [39]. Another argument is that failure in decision-making or risk management, caused by insufficient assessment of potential risk, may increase the risk of injuries [40]. One included study [30] also considered the previous injury as a strong risk factor for subsequent injury. Li et al. [26] found that depressive symptoms are only a risk factor for injury in men when paired with anxiety. Despite the fact that one study [23] found that women exhibited greater depressive symptoms than men, Belz et al. [24] and Li et al. [26] found no gender differences in the risk of depression among injured athletes. Notwithstanding the controversial results, these findings were expected since the literature pointed out that depressive symptoms are higher among women in the general population [41], and specifically among athletes [14]. Only one study assessed the prevalence of depressive symptoms among collegiate athletes, being 21.7% [26], lower than was found in other studies [13,14].

Despite some intriguing findings, the current study had some limitations that must be addressed. First, the studies included in our analysis involved a variety of sports and MSK injuries (type and location), thereby hindering our ability to perform analyses based on specific sports or types/locations of injuries. Moreover, most studies include a sample of young athletes, so more research with senior athletes is needed to explore the association between MSK injuries and depressive symptoms in more detail. Finally, this review includes five low-quality methodological studies (three cross-sectional studies and two cohort studies). Therefore, high-quality studies and longitudinal studies are mandatory to further explore the relationship and the cause-effect between MSK injuries and depression.

## 5. Conclusions

The findings of this review verified a bi-directional association between the presence of MSK injuries caused by sports participation and the presence of depressive symptoms in athletes. Although psychological status is often recognized as a contributor to poor performance and high levels of injury, it is rarely considered a factor in injury prevention. Similarly, it is vital to acknowledge that MSK injuries may increase the risk of developing depressive symptoms. One practical implication of this review is the awareness of depressive symptoms as a risk factor for musculoskeletal injuries, as well as the need for an athlete to obtain psychological assistance following an injury. Thus, future studies could explore the cause-effect relationship between depressive symptoms and MSK injury.

## Figures and Tables

**Figure 1 ijerph-20-06130-f001:**
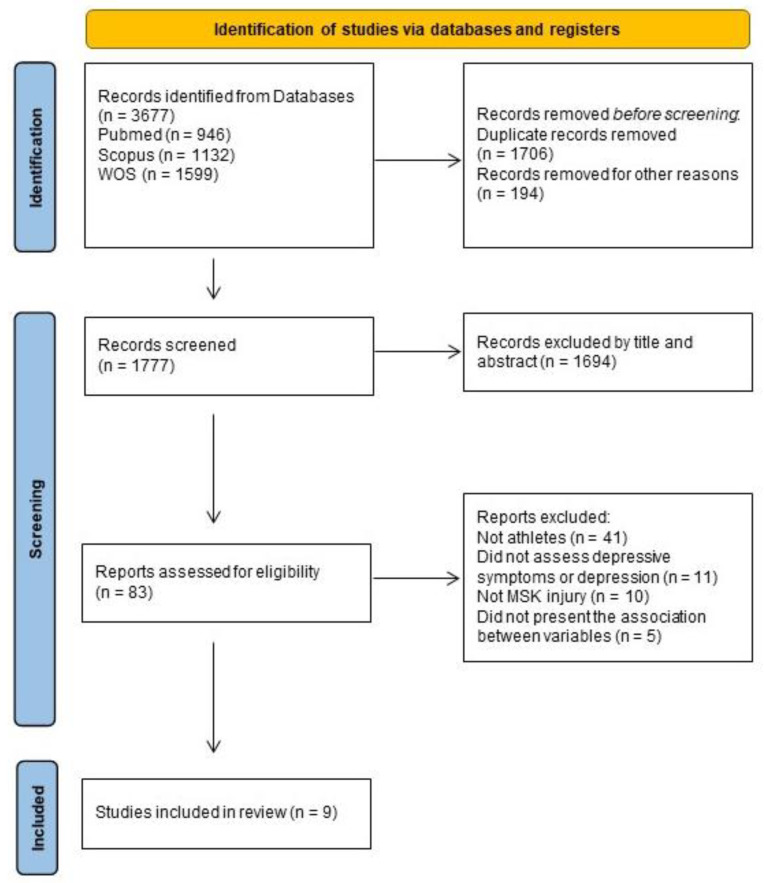
PRISMA flowchart.

**Table 1 ijerph-20-06130-t001:** Characteristics of included studies.

First Author, Year, Country	Design	Participants Characteristics	Sport Modality and Level	Depressive Symptoms Assessment	MSK Injury	Main Results
Appaneal, 2009 [23], USA	Prospective Cohort Study	164 student-athletes (108 men), mean age 19.7 ± 2.0	46% football, 17% basketball, 15% soccer, 9% volleyball, 6% baseball, 4% gymnastics, 3% track and field, and 1% wrestling 71% starters athletes and 85% collegiate level	CES-D and POMS	Joints (ligament and dislocations), muscle, bone (fractures), head (concussions) and neck (cervical spine), low back (disc)	Athletes with injuries had higher depression rates than those without injuries after one week and remained elevated for up to one month. Women had greater depression symptom severity than men. Men had more severe injuries than women.
Belz, 2018 [24], Germany	Cross-section design	154 competitive athletes (49% men), mean age 18.7 ± 5.0	No information	PHQ-2	Back pain	Risk for depression was a significant predictor of disability among athletes with back pain.
Lavallée, 1996 [25], Canada	Cross-section design	55 athletes, only men, mean age 22	42 football, 13 rugby	POMS	Did not specify	Among rugby athletes, depression/dejection was related to injury frequency.
Li, 2017 [26], USA	Prospective Cohort Study	958 Collegiate athletes (65.9% men), aged between 18 and 22	Male athletes (i.e., baseball, basketball, football, and wrestling) Female athletes (i.e., basketball, field hockey, softball, football, and volleyball)	20-item CES-D	Sprain (49.1%), strain (13.1%), fracture (9.5%), concussion (9.0%)	The prevalence of depression at the preseason was 21.7%. The injury risk did not increase among female athletes nor in male athletes who reported preseason depressive symptoms but not in the presence of anxiety symptoms. Anxiety combined with depressive symptoms increased the risk of injury in men.
San-Antolin, 2020 [27], Spain	Secondary Case-Control	50 athletes (25 case group, 25 control group), 27 men, mean age 41 ± 11.7, case group and 38 ± 11.8 control group	Football, athletics, running, fitness, and paddle.	BDI-II	Myofascial pain (n = 25), healthy athletes (n = 25)	Greater depression symptoms in athletes with gastrocnemius myofascial pain versus healthy athletes.
Smith, 1993 [28], USA	Prospective Cohort Study	276 athletes (238 men), no age information	Hockey, basketball, baseball, and volleyball teams	POMS	Did not specify	Significant increments were noted among depression symptoms from preinjury to postinjury stages.
Tomlinson, 2021 [29], USA	Prospective observational study	51 collegiate athletes (22 men), mean age 20.0 ± 1.4	Football and cross-country	CES-D	Fracture	Injured athletes showed fewer depressive symptoms compared to non-injured athletes.
Yang, 2014 [30], USA	Prospective Cohort Study	330 university athletes (only men), no age information	Football	CES-D	Knee injury (22.35%) most common region, and sprain most common type (46.3%)	Depression was associated with a greater likelihood of injury. Football players who experienced depression at baseline were 10% less likely to remain injury-free than those who did not have depressive symptoms.
Gouttebarg, 2016 [31], five countries in Europe	Cross-sectional	540 athletes (men), mean age 26.7 ± 4.4	Football 54% playing in the highest leagues	GHQ-12	Did not specify	Professional footballers who had sustained one or more severe MSK injuries during their career were two to nearly four times more likely to report symptoms of common mental disorders than professional footballers who had not suffered from severe MSK injuries.

Abbreviation: MSK, musculoskeletal; CES-D, Center for Epidemiologic Studies Depression; POMS, Profile of Mood States; PHQ-2, Patient-Health- Questionnaire-2; BDI-II Beck Depression Inventory-II; GHQ-12, 12-item General Health Questionnaire.

**Table 2 ijerph-20-06130-t002:** The methodological quality of cohort studies.

Study	Selection				Comparability		Exposure			Total Score
	1	2	3	4	5	6	7	8	9	
Appaneal et al., 2009 [23]	+	+	+	+	+	−	+	−	−	6
Li et al., 2017 [26]	+	+	−	+	+	+	−	+	−	6
Smith et al., 1993 [28]	+	+	−	+	−	−	−	−	−	3
Tomlinson et al., 2021 [29]	−	+	−	+	−	−	−	−	−	2
Yang et al., 2014 [30]	+	+	−	+	+	+	−	+	+	7

Newcastle-Ottawa quality assessment scale: Legend: +, score fulfilled; −, score not fulfilled/too little information; 1, representativeness of the exposed cohort; 2, selection of the non-exposed cohort; 3, ascertainment of exposure; 4, the demonstration that outcome of interest was not present at the start of the study; 5 and 6 comparabilities of cohorts based on the design or analysis; 7, assessment of outcome; 8, was follow-up long enough for outcomes to occur; 9, adequacy of follow-up of cohorts.

**Table 3 ijerph-20-06130-t003:** The methodological quality of the case-control study.

Study	Selection				Comparability		Exposure			Total Score
	1	2	3	4	5	6	7	8	9	
San-Antolin et al., 2020 [28]	−	+	+	+	+	+	−	+	−	6

Newcastle-Ottawa quality assessment scale: Legend: +, score fulfilled; −, score not fulfilled/too little information; 1, Is the case definition adequate; 2, representative of the cases; 3, selection of controls; 4, the definition of controls; 5 and 6, comparability of cases and controls based on the design analysis; 7, ascertainment of exposure; 8, the same method of ascertainment for cases and controls; 9, non-response rate.

**Table 4 ijerph-20-06130-t004:** The methodological quality of cross-sectional studies.

Study	Selection				Comparability		Outcome		Total Score
	1	2	3	4	5	6	7	8	
Belz et al., 2018 [24]	+	−	−	−	+	−	−	+	3
Lavallée et al., 1996 [25]	−	−	−	−	−	−	−	−	0
Gouttebarge et al., 2016 [31]	+	+	−	−	−	−	−	+	3

Newcastle-Ottawa quality assessment scale: Legend: +, score fulfilled; −, score not fulfilled/too little information; 1, representative of the sample; 2, sample size; 3, non-respondents; 4, ascertainment of the exposure; 5 and 6; comparability of subjects in different outcome groups based on design or analysis; 7, assessment of outcome; 8, statistical test.

## Data Availability

Please contact the corresponding author to discuss the availability of the data and materials.

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
