# Peer review of "Association between Musculoskeletal Injuries and Depressive Symptoms among Athletes: A Systematic Review"

_ijerph, 2023, doi:10.3390/ijerph20126130_

Round 1

Reviewer 1 Report

Thank you for the opportunity to review this article.

The summary is adequately structured and includes all the points necessary for understanding the study.

Introduction: the authors adequately present the theme using current and relevant bibliography, however, they should present the research question and the objective of the present study more clearly.

Materials and methods: the authors present the research question in this section, however, it seems more pertinent that it appears in the introduction so that the reader can perceive the motivations for the study.

In the research strategy, did the authors not mention a time limit, did they not apply any limits? This is important to mention.

In this section, it is important that the authors define the PICO question, since this is a systematic review. The inclusion and exclusion criteria should also be better defined, referencing the type of studies that were included, for example: experimental, quasi-experimental, among others.

Regarding search terms, did you use the same terms for all databases? They only used the terms MESH, why not use free terms in the title and abstract?

The results are adequately described, however, in section 3.8 there is an error in referencing the tables, it is not 1-3, but 3-5.

Discussion: the authors discuss the results properly confronting the literature.

Conclusion: here the authors must give a clear answer to the objective of the study and will be able to address the implications of this study for practice, for academia and for research.

References must be reviewed, as they are not in accordance with the journal's guidelines.

These are in adequate number, although there are articles with more than 10 years, these were the results of the search. Another literature used with more than 10 years seems adequate and pertinent.

Author Response

Thank you for the opportunity to revise our manuscript and resubmit it to International Journal of Environmental Research and Public Health.

We are very grateful to the reviewers for their overall very positive evaluation and their detailed and helpful suggestions for improvement. In a thorough revision, we have now addressed all of the comments raised by the reviewers and we believe that the manuscript has substantially improved.

We hope the revised version is now suitable for publication in International Journal of Environmental Research and Public Health.

Yours sincerely,

Dr. Priscila Marconcin

Reviewer 1

Comment: The summary is adequately structured and includes all the points necessary for understanding the study. Introduction: the authors adequately present the theme using current and relevant bibliography, however, they should present the research question and the objective of the present study more clearly.

Answer: Thank you for your positive evaluation. We put the research question in the introduction section.

Comment: Materials and methods: the authors present the research question in this section, however, it seems more pertinent that it appears in the introduction so that the reader can perceive the motivations for the study.

 Answer: Thank you, we did it.

Comment: In the research strategy, did the authors not mention a time limit, did they not apply any limits? This is important to mention.

 Answer: No start date has been set, all publication until 15 February 2023 were included. We clarity this point.

Comment: In this section, it is important that the authors define the PICO question, since this is a systematic review. The inclusion and exclusion criteria should also be better defined, referencing the type of studies that were included, for example: experimental, quasi-experimental, among others.

 Answer: Thank you for your comment. We did not used PICO question because we did not search experimental studies, in this sense do not make sense to describe an intervention or a comparison. We describe the population and the outcome. And in the line 99 and 100 we describe the study designs required

Comment: Regarding search terms, did you use the same terms for all databases? They only used the terms MESH, why not use free terms in the title and abstract?

 Answer: Thank you for your comment. We used the same terms for all databases, and we add this information in the text. We opted to use MESH terms because it impose uniformity and consistency for the search.

Comment: The results are adequately described, however, in section 3.8 there is an error in referencing the tables, it is not 1-3, but 3-5.

 Answer: Thank you. We corrected it.

Comment: Discussion: the authors discuss the results properly confronting the literature.

 Answer: Thank you.

Comment: Conclusion: here the authors must give a clear answer to the objective of the study and will be able to address the implications of this study for practice, for academia and for research.

Answer: Thank you. We add the practical implication and rewrite the conclusion section.

Comment: References must be reviewed, as they are not in accordance with the journal's guidelines.

Answer: Thank you, we reviewed the reference list according to MDPI instructs.

Comment: These are in adequate number, although there are articles with more than 10 years, these were the results of the search. Another literature used with more than 10 years seems adequate and pertinent.

Answer: Thank you for the suggestions. In fact, we also followed other reviewers’ comments and improve the quality of the whole section, indicating the missing references.

Reviewer 2 Report

The authors provide a systematic review concerning the effects of musculoskeletal injuries on depressive symptoms in athletes, also researching the other direction of impact. Using PRISMA guidelines and applying the Newcastle-Ottawa Scale (NOS) the review revealed a bidirectional association between musculoskeletal injuries and depression. The review depicts interesting gender differences. The authors conclude that both directions of impact (injury, depressive symptoms) should be considered in training routines.

The review is well presented, meets PRISMA guidelines – except the report of effect sizes, and applies a new classification guideline for non-randomized studies. The review was carefully conducted. The article provides important information concerning selected articles in representative tables. The lack of information according effect sizes (forest plots) should be explained. Some English expressions have to be revised for solid understanding.

Comments in detail:

Abstract: MSK injuries have …  
line 17: date of inception? should rather be the date of termination?

line 46: American football or soccer?
line 49: … population, and is a ….
line 51: difficulty of?....
line 55: Referring to the same investigation, female college athletes…
line 85: a third researcher was asked ….
line 92: a sample … or: samples ….
line 95:…  or (a) cross sectional study design
line 101:  results were extracted
line 112, 113: were rated as low-quality studies….
(line 136: a case-control study … or … case controlled)

table 1: Li et al.: the injury risk did not increase among ….
line 144: , and the Smith et al. study
line 147: age of 41 years for the case group and 38 years for the control group
line 151: … and a minority came from individual ….

line 153: I would rather say “sport experience”, in line 156 you may use … the participants’ levels of sport performance

line 172: another study (or: other studies) displayed
line 192: conclusions came from …
line 195: , and this was in line with the majority of evidence… ?
line 201: low-quality-studies

Discussion: line 223 – 227: the text in the present form would need references. The summary of evidence could be stated more clearly: you might combine [29] and [36] to one outcome etc. Describe the statement and then refer to the studies (or even one study – high or low quality…)

line 225: to leverage his/her athletic performance  
line 227: symptoms substantially increase an athlete’s chance to fail (would need a reference or you should rephrase to: could, might)
line 231: … and this is in agreement …?
line 241: the feeling of identity loss or anger that came from ….
line 255: symptoms are risk factors …. or: symptoms are a risk factor
line 261: than it was found in other studies …
line 266: studies with senior athletes are needed …
line 269: please rephrase the last two sentences of the discussion to achieve better understanding – mandatory to further explore?...

Conclusion: line 277: please check this sentence – should it be linked to the sentence before?

please check the format of the reference list: I think MDPI instructs to name all authors of a paper

Author Response

Thank you for the opportunity to revise our manuscript and resubmit it to International Journal of Environmental Research and Public Health.

We are very grateful to the reviewers for their overall very positive evaluation and their detailed and helpful suggestions for improvement. In a thorough revision, we have now addressed all of the comments raised by the reviewers and we believe that the manuscript has substantially improved.

We hope the revised version is now suitable for publication in International Journal of Environmental Research and Public Health.

Yours sincerely,

Dr. Priscila Marconcin

Comment: The authors provide a systematic review concerning the effects of musculoskeletal injuries on depressive symptoms in athletes, also researching the other direction of impact. Using PRISMA guidelines and applying the Newcastle-Ottawa Scale (NOS) the review revealed a bidirectional association between musculoskeletal injuries and depression. The review depicts interesting gender differences. The authors conclude that both directions of impact (injury, depressive symptoms) should be considered in training routines.

Answer: Thank you for reviewing our study and for all your comments.

Comment: The review is well presented, meets PRISMA guidelines – except the report of effect sizes, and applies a new classification guideline for non-randomized studies. The review was carefully conducted. The article provides important information concerning selected articles in representative tables. The lack of information according effect sizes (forest plots) should be explained. Some English expressions have to be revised for solid understanding.

Answer: Thank you. We did not the forest plots because the studies did not have the same outcome. We did an English review.

Comments in detail:

Comment: Abstract: MSK injuries have … 

Answer: Thank you, we did the correction.

line 17: date of inception? should rather be the date of termination?

Answer: Thank you. We rewrite the sentence, we do not set the start date of the surveys.

Comment: line 46: American football or soccer?

Answer: American Football.

Comment: line 49: … population, and is a ….

Answer: Thank you, we did the correction.

Comment: line 51: difficulty of?....

Answer: Thank you, we did the correction.

Comment: line 55: Referring to the same investigation, female college athletes…

Answer: Thank you, we did the correction.

Comment: line 85: a third researcher was asked ….

Answer: Thank you, we did the correction.

Comment: line 92: a sample … or: samples ….

Answer: Thank you, we did the correction.

Comment: line 95:…  or (a) cross sectional study design

Answer: Thank you, we did the correction.

Comment: line 101:  results were extracted

Answer: Thank you, we did the correction.

Comment: line 112, 113: were rated as low-quality studies….

Answer: Thank you, we did the correction.

Comment: (line 136: a case-control study … or … case controlled)

Answer: Thank you, we did the correction.

Comment: table 1: Li et al.: the injury risk did not increase among ….

Answer: Thank you, we did the correction.

Comment: line 144: , and the Smith et al. study

Answer: Thank you, we did the correction.

Comment: line 147: age of 41 years for the case group and 38 years for the control group

Answer: Thank you, we did the correction.

Comment: line 151: … and a minority came from individual ….

Answer: Thank you, we did the correction.

Comment: line 153: I would rather say “sport experience”, in line 156 you may use … the participants’ levels of sport performance

Answer: Thank you, we did the correction.

Comment: line 172: another study (or: other studies) displayed

Answer: Thank you, we did the correction.

Comment: line 192: conclusions came from …

Answer: Thank you, we did the correction.

Comment: line 195: , and this was in line with the majority of evidence… ?

Answer: Thank you, we did the correction.

Comment: line 201: low-quality-studies

Answer: Thank you, we did the correction.

Comment: Discussion: line 223 – 227: the text in the present form would need references. The summary of evidence could be stated more clearly: you might combine [29] and [36] to one outcome etc. Describe the statement and then refer to the studies (or even one study – high or low quality…)

Answer: Thank you for the suggestions. In fact, we also followed other reviewers’ comments and improve the quality of the whole section, indicating the missing references.

Comment: line 225: to leverage his/her athletic performance 

Answer: Thank you, we did the correction.

Comment: line 227: symptoms substantially increase an athlete’s chance to fail (would need a reference or you should rephrase to: could, might)

Answer: Thank you, we did the correction.

Comment: line 231: … and this is in agreement …?

Answer: Thank you, we did the correction.

Comment: line 241: the feeling of identity loss or anger that came from ….

Answer: Thank you, we did the correction.

Comment: line 255: symptoms are risk factors …. or: symptoms are a risk factor

Answer: Thank you, we did the correction.

Comment: line 261: than it was found in other studies …

Answer: Thank you, we did the correction.

Comment: line 266: studies with senior athletes are needed …

Answer: Thank you, we did the correction.

Comment: line 269: please rephrase the last two sentences of the discussion to achieve better understanding – mandatory to further explore?...

Answer: Thank you, we rewrite the last sentence.

Comment: Conclusion: line 277: please check this sentence – should it be linked to the sentence before?

Answer: Thank you, it will be better in just one paragraph.

Comment: please check the format of the reference list: I think MDPI instructs to name all authors of a paper

Answer: Thank you, we reviewed the reference list according to MDPI instructs.

Reviewer 3 Report

The innovation of this study is limited. 

"abstract" with grammatical errors.

Literature Search Strategy is not well described.

Some reasonable adjustments need to be made

I suggest a deep reading of the study for corrections and improvement

The manuscript needs revision for language and grammar.

Author Response

Thank you for the opportunity to revise our manuscript and resubmit it to International Journal of Environmental Research and Public Health.

We are very grateful to the reviewers for their overall very positive evaluation and their detailed and helpful suggestions for improvement. In a thorough revision, we have now addressed all of the comments raised by the reviewers and we believe that the manuscript has substantially improved.

We hope the revised version is now suitable for publication in International Journal of Environmental Research and Public Health.

Yours sincerely,

Dr. Priscila Marconcin

Reviewer 3

Comment: The innovation of this study is limited. 

"abstract" with grammatical errors.

Literature Search Strategy is not well described.

Some reasonable adjustments need to be made

I suggest a deep reading of the study for corrections and improvement

Answer: Thank you for your comments. We have revised the manuscript based on the detailed comments from the reviewers. We have also reviewed the English language throughout the article. As a result, we hope that the article has improved enough to be accepted for publication in IJERPH. We acknowledge the study's limitations, but we believe it can be a valuable contribution as a starting point for further exploration of the relationship between musculoskeletal injury and mental health, particularly among athletes.

Reviewer 4 Report

This study is a systematic review on the association between musculoskeletal (MSK) injuries with depressive symptoms among athletes.  After screening based on inclusion and exclusion criteria, the authors included nine studies in the review.  They drew several conclusions: that there is a bidirectional association between MSK injuries and depressive symptoms; that athletes with MSK injuries had higher levels of depressive symptoms, which in turn increases the risk of depression; that women athletes had higher levels of depressive symptoms compared to men athletes; that depression predicted disability.   

Major comment:

Although the topic is of potential significance for those in sports medicine and psychology, the evidence from the nine studies for drawing these conclusions about complex associations and mechanisms as stated (e.g. bidirectional relationships, mediation, gender differences) is slim and far from sufficient.  Drawing conclusions about these complex associations and mechanisms would require several systematic reviews that address the particular associations and mechanisms that involve the constructs or variables in question. I am uncertain if this flaw can be fixed even with a major revision of the manuscript. 

Other comments:

The reasons for excluding 1694 records at screening are unclear. (Section 3.1 and Figure 1) Please provide the reasons.

- Did all studies use the 20-item CESD scale? Overall, the number of items of the different scales for assessing depressive symptoms was provided for some studies but not others in Table 1. Table 1 requires greater reporting consistency.

- “Date of inception” in the abstract does not make sense. Does that refer to the end date of the search

- Some wordings are problematic—e.g. “Athletes’ women” should be “Women athletes” (line 21); the meaning of "post-MSK injuries" is unclear (line 93).  Proofreading of the whole manuscript is strong recommended. 

Professional editing would be required-- overall it reads fine; however, some terms and expressions were used incorrectly and potentially misleading.

Author Response

Thank you for the opportunity to revise our manuscript and resubmit it to International Journal of Environmental Research and Public Health.

We are very grateful to the reviewers for their overall very positive evaluation and their detailed and helpful suggestions for improvement. In a thorough revision, we have now addressed all of the comments raised by the reviewers and we believe that the manuscript has substantially improved.

We hope the revised version is now suitable for publication in International Journal of Environmental Research and Public Health.

Yours sincerely,

Dr. Priscila Marconcin

Comment: This study is a systematic review on the association between musculoskeletal (MSK) injuries with depressive symptoms among athletes.  After screening based on inclusion and exclusion criteria, the authors included nine studies in the review.  They drew several conclusions: that there is a bidirectional association between MSK injuries and depressive symptoms; that athletes with MSK injuries had higher levels of depressive symptoms, which in turn increases the risk of depression; that women athletes had higher levels of depressive symptoms compared to men athletes; that depression predicted disability.   

Major comment:

Although the topic is of potential significance for those in sports medicine and psychology, the evidence from the nine studies for drawing these conclusions about complex associations and mechanisms as stated (e.g. bidirectional relationships, mediation, gender differences) is slim and far from sufficient.  Drawing conclusions about these complex associations and mechanisms would require several systematic reviews that address the particular associations and mechanisms that involve the constructs or variables in question. I am uncertain if this flaw can be fixed even with a major revision of the manuscript. 

Answer: Thank you for your comments. We have revised the manuscript based on the detailed comments from the reviewers. As a result, we hope that the article has improved enough to be accepted for publication in IJERPH. We acknowledge the study's limitations, but we believe it can be a valuable contribution as a starting point for further exploration of the relationship between musculoskeletal injury and mental health, particularly among athletes.

Other comments:

Comment: -  The reasons for excluding 1694 records at screening are unclear. (Section 3.1 and Figure 1) Please provide the reasons.

Answer: Thank you. We review the study selection and correct it, both in the text and in the PRISMA flowchart.

Comment: - Did all studies use the 20-item CESD scale? Overall, the number of items of the different scales for assessing depressive symptoms was provided for some studies but not others in Table 1. Table 1 requires greater reporting consistency.

Answer: Thank you. We described in table 1 each instrument that assessed depressive symptoms, sucha as BDI-II, GHQ-12 and others.

Comment: - “Date of inception” in the abstract does not make sense. Does that refer to the end date of the search

Answer: Thank you. We correct it.

Comment: - Some wordings are problematic—e.g. “Athletes’ women” should be “Women athletes” (line 21); the meaning of "post-MSK injuries" is unclear (line 93).  

Answer: Thank you. We correct it.

Comment: Proofreading of the whole manuscript is strong recommended. 

Answer: Thank you.

Round 2

Reviewer 2 Report

The authors provide a revised version of their systematic review concerning musculoskeletal injuries and depressive symptoms. The inserted text supports the flow of reading. Although the text was substantially improved, some minor comments should be considered.

Abstract,
in line 25: sports managers are more involved in financial and advertising activities supporting the athlete. To my opinion, “coaches” would fit better.

Introduction,
in line 44: the relation is missing (five basketball players per 10,000??)
line 53: difficulty of thinking

throughout the text, please use the same term: depressive symptoms (e.g. depressed symptoms in line 56, in line 72…)

line 68: delete that….. depressive symptoms

PRISMA flowchart: screened records minus excluded records by title and abstract would be 83

table 1: 46% football and - referring to the same study - 15% football cannot be correct. So please be careful in the distinction of American football and soccer.
those without injuries “at one week”: after one week

Li et al. 2017: Sprain, Strain, fracture, concussion… stay with capital letters or no capital letters

line 174: and another study…
line 180: fractures
line 196: The author’s findings
line 210, 217, 223: Newcastle-Ottawa quality assessment scale (delete “a”)
line 216: Table 4: The methodological
line 272: more research … is needed to explore … in more detail…
line 273: this review includes….

Author Response

The authors provide a revised version of their systematic review concerning musculoskeletal injuries and depressive symptoms. The inserted text supports the flow of reading. Although the text was substantially improved, some minor comments should be considered.

 Abstract,
in line 25: sports managers are more involved in financial and advertising activities supporting the athlete. To my opinion, “coaches” would fit better.

Answer: Thank you. We corrected that.

Introduction,
in line 44: the relation is missing (five basketball players per 10,000??)
line 53: difficulty of thinking

Answer: Thank you. We corrected that.

throughout the text, please use the same term: depressive symptoms (e.g. depressed symptoms in line 56, in line 72…)

Answer: Thank you. We corrected that.

line 68: delete that….. depressive symptoms

Answer: Thank you. We corrected that.

PRISMA flowchart: screened records minus excluded records by title and abstract would be 83

Answer: Thank you. We corrected that.

table 1: 46% football and - referring to the same study - 15% football cannot be correct. So please be careful in the distinction of American football and soccer.
those without injuries “at one week”: after one week

Answer: Thank you. We corrected that.

Li et al. 2017: Sprain, Strain, fracture, concussion… stay with capital letters or no capital letters

Answer: Thank you. We corrected that.

line 174: and another study…
line 180: fractures
line 196: The author’s findings
line 210, 217, 223: Newcastle-Ottawa quality assessment scale (delete “a”)
line 216: Table 4: The methodological
line 272: more research … is needed to explore … in more detail…
line 273: this review includes….

Answer: Thank you. We corrected all that.

Reviewer 3 Report

Dear Authors,

Congratulations on the review the study, has improved significantly.

I suggest a deep reading to correct any errors, as in line 18.

line 18: We search or We searched

The quality of English has improved, however I suggest reading it to correct any errors

Author Response

Dear Authors,

Congratulations on the review the study, has improved significantly.

I suggest a deep reading to correct any errors, as in line 18.

line 18: We search or We searched

Answer: Thank you. We corrected this mistake, and did a deep reading again.